# Association between Periodontitis and Hematologic Cancer: An NHIRD Cohort Study in Taiwan

**DOI:** 10.3390/cancers16091671

**Published:** 2024-04-25

**Authors:** Liang-Gie Huang, Cheng-Chia Yu, Ming-Ching Lin, Yu-Hsun Wang, Yu-Chao Chang

**Affiliations:** 1School of Dentistry, Chung Shan Medical University, Taichung 40201, Taiwan; lianggie.huang@gmail.com; 2Department of Stomatology, Taichung Veterans General Hospital, Taichung 40705, Taiwan; mclin425@gmail.com; 3Department of Industrial Engineering and Enterprise Information, Tunghai University, Taichung 40704, Taiwan; 4Institute of Oral Sciences, Chung Shan Medical University, Taichung 40201, Taiwan; 5Department of Medical Research, Chung Shan Medical University Hospital, Taichung 40201, Taiwan; cshe731@csh.org.tw; 6Department of Dentistry, Chung Shan Medical University Hospital, Taichung 40201, Taiwan

**Keywords:** chronic periodontitis, hematologic cancers, leukemia, lymphoma

## Abstract

**Simple Summary:**

Our study aimed to delineate the variables associated with the link between chronic periodontitis and hematologic cancers. The hazard ratio for hematologic cancers in patients with chronic periodontitis was also evaluated. Comprehensive statistical analyses revealed a 1.25-fold risk of hematologic cancers in the chronic periodontitis group. Factors such as being male and having hypertension were identified as increased risk factors for hematologic cancers. This nuanced exploration, including a subtype analysis for leukemia and lymphoma, contributes valuable insights into the complex relationship between chronic periodontitis and specific hematologic cancer subtypes. These findings could enhance our understanding of potential cancer risk factors in Taiwan.

**Abstract:**

Background: Chronic periodontitis, an inflammation-related disorder affecting global populations, has been revealed to be linked to diverse cancers. Numerous epidemiological studies have not shown a link between chronic periodontitis and blood cancers in Taiwan. Methods: This study included 601,628 patients, diagnosed with newly chronic periodontitis by the ICD-9-CM classification, who were enrolled from 2001 to 2021 in the National Health Insurance Research Database (NHIRD) in Taiwan. In this study, we employed comprehensive statistical analyses to investigate the association between chronic periodontitis and hematologic cancers. Initially, we calculated incidence density and used a Poisson regression to analyze relative risk. Subsequently, we compared the cumulative incidence of hematological cancer in both chronic and non-chronic periodontitis groups using the Kaplan–Meier method. Results: The results revealed a significantly lower cumulative incidence of hematologic cancer in individuals with non-chronic periodontitis over a 12-year follow-up period. To further explore the risk factors, a Cox proportional hazard regression analysis was conducted. Being male (adjusted hazard ratio [aHR] = 1.21, 95% CI: 1.04 to 1.42; *p* = 0.014) and having hypertension (aHR = 1.34, 95% CI: 1.06 to 1.69; *p* = 0.015) were demonstrated to be associated with an increased risk of hematologic cancers, respectively. In addition, in a subtype multivariate analysis for categorizing hematologic cancers into lymphoma and leukemia, the aHR for leukemia was 1.48 (95% CI: 1.13 to 1.93; *p* = 0.004) and aHR for lymphoma was 1.15 (95% CI: 0.96 to 1.37; *p* = 0.140). Conclusions: This study found that being male and having hypertension were the significant risk factors for hematological malignancies. Moreover, the association between chronic periodontitis and specific subtypes of hematologic cancers was confirmed.

## 1. Introduction

Chronic periodontitis is an inflammation-related disorder affecting the periodontal tissue, triggered by an imbalance in the oral microbial biofilm. Periodontal disease is prevalent worldwide, affecting 20–50% of the global population, with approximately 10% of people suffering from severe periodontitis [1,2,3]. Recently, significant endeavors have been undertaken to clarify the impact of an imbalanced oral microbiome on diverse systemic conditions, such as cardiovascular disease and cancer [4,5,6]. Epidemiological studies have identified a positive association between chronic periodontitis and the incidence of diverse cancer types, including head and neck cancer, esophageal cancer, gastric cancer, pancreatic cancer, colorectal cancer, lung cancer, breast cancer, gall bladder cancer, liver cancer, prostate cancer, hematological/hematopoietic malignancies, and genitourinary cancers [7,8,9,10,11].

Hematologic cancer, also known as hematological malignancy or blood cancer, refers to cancers that affect the blood, bone marrow, lymphatic system, and lymphoid tissues [12,13]. These cancers originate in the cells of the hematopoietic and lymphoid systems, which are responsible for the production of blood cells and the immune system [14]. The main types of hematologic cancers include (1) Leukemia: A type of cancer that begins in blood-forming tissues, such as bone marrow, and results in the excessive production of abnormal white blood cells. Leukemia can be manifested as an acute or chronic condition. (2) Lymphoma: Cancer that starts in the lymphocytes, a type of white blood cell, and primarily affects the lymph nodes and lymphoid tissues. There are two main types, Hodgkin lymphoma and non-Hodgkin lymphoma [15]. Hematologic cancers can interfere with the normal function of blood cells, weaken the immune system, and cause various symptoms such as fatigue, anemia, frequent infections, and abnormal bleeding [16]. Treatment for hematologic cancers may include chemotherapy, radiation therapy, immunotherapy, targeted therapy, and stem cell transplantation, depending on the specific type and stage of the cancer [17].

Until present, multiple epidemiological studies have investigated the potential risk of hematopoietic cancers in individuals with periodontitis [9,10,18,19,20,21,22]. Nevertheless, these studies have yielded inconsistent results, and the evidence still remains inconclusive. Therefore, we established two objectives to explore in this research: (1) to delineate the variables associated with the connection between chronic periodontitis and hematologic cancers; (2) to evaluate whether a higher hazard ratio for hematologic cancers existed within the group of patients with chronic periodontitis.

## 2. Materials and Methods

### 2.1. Data Sources, Study Population, and Participant Selection

Data were sourced from the National Health Insurance Research Database (NHIRD) in Taiwan. The National Health Research Institute initiated its National Health Insurance Program in 1995, aiming to enhance healthcare for the entire Taiwanese population. In terms of the current study, participants newly diagnosed with periodontitis were enrolled from 2001 to 2021 if they had attended dental outpatient visits three times or more. Participants with no history of diagnosed periodontitis were included, from 2000 to 2013. In the subset of those with newly diagnosed periodontitis, individuals with a diagnosed cancer prior to the index date were excluded. For both groups, we implemented 1:1 matching based on age, sex, monthly income, urbanization, and comorbidities.

### 2.2. The Definitions of Chronic Periodontitis and Hematologic Cancer

The identification of periodontitis was captured using ≥2 outpatient visits for dental treatment based on the ICD-9-CM diagnostic codes 523.4 and 523.5 from 2001 to 2012. Individuals with hematologic cancer were defined as those with ≥2 outpatient visits or 1 hospitalization. Moreover, the ICD-9-CM diagnostic codes for lymphoma and leukemia are 200–203 and 204–208, respectively. In the assessment of independent variables, conditions including hypertension (ICD-9-CM codes: 401–405), hyperlipidemia (ICD-9-CM codes: 272.0–272.4), diabetes mellitus (ICD-9-CM code: 250), chronic obstructive pulmonary disease (ICD-9-CM codes: 491, 492, 496), thyroid disease (ICD-9-CM codes: 240–246), asthma (ICD-9-CM code: 493), myocardial infarction (ICD-9-CM codes: 410–414), stroke (ICD-9-CM codes: 430–438), and insomnia (ICD-9-CM code: 780.52) were considered. The observation period was the year following the starting point at which individuals had been diagnosed through at least 2 outpatient visits or 1 inpatient diagnosis. The ICD-9-CM codes 401, 402, 403, 404, and 405 denote essential hypertension, hypertensive heart disease, hypertensive chronic kidney disease, hypertensive heart and chronic kidney disease, and secondary hypertension, respectively. The ICD-9-CM codes 272.0, 272.1, 272.2, 272.3, and 272.4 correspond to pure hypercholesterolemia, pure hyperglyceridemia, mixed hyperlipidemia, hyperchylomicronemia, and other and unspecified hyperlipidemia, respectively.

### 2.3. Independent Variable Assessment

We evaluated various independent variables that could be associated with both chronic periodontitis and hematologic cancer. These independent variables included age (<18, 18–64, ≥65 years old), sex (female and male), monthly income (<NTD 25,000, NTD 25,000, NTD 40,000, >NTD 40,000), urbanization (urban, suburban, rural), hypertension (no or yes), hyperlipidemia (no or yes), diabetes (no or yes), chronic obstructive pulmonary disease (no or yes), thyroid disease (no or yes), asthma (no or yes), myocardial infarction (no or yes), stroke (no or yes), and insomnia (no or yes). 

### 2.4. Statistical Analyses

All statistical analyses were performed using SAS version 9.4 software (SAS Institute, Cary, NC, USA). All data values are presented as the mean ± standard error. The variables of demographic characteristics were used to determine the difference between chronic periodontitis and non-chronic periodontitis via Chi-squared tests. Poisson regression analysis was used to verify relative risk of hematologic cancers between chronic periodontitis and non-chronic periodontitis. A Kaplan–Meier analysis was employed to illustrate the cumulative incidence of hematologic cancers between chronic periodontitis and non-chronic periodontitis. Cox proportional hazard models were utilized to calculate the hazard ratios and 95% confidence intervals for hematologic cancers. Differences were considered significant at a *p* value < 0.05.

## 3. Results

A total of 601,628 individuals were sampled from the longitudinal health insurance database and categorized into a newly diagnosed periodontitis group (n = 255,052) and a never-diagnosed periodontitis group (n = 346,576), as illustrated in Figure 1. In the newly diagnosed periodontitis group (n = 254,765), those diagnosed with cancer before the index date was also excluded. Prior to propensity score matching, both the newly diagnosed periodontitis group (n = 190,455) and the never-diagnosed periodontitis group (n = 190,455) were matched at a 1:1 ratio based on age and sex (Figure 1). In order to mitigate bias in examining the connections between chronic periodontitis and the outcomes of hematologic cancers, propensity score matching was used to adjust the potential confounding factors. Finally, the periodontitis and non-periodontitis groups included 168,191 and 168,191 individuals, via propensity score matching, for further analysis (Figure 1).

As shown in Table 1, no significant differences were revealed in age, sex, monthly income, urbanization, hypertension, hyperlipidemia, diabetes mellitus, chronic obstructive pulmonary disease, myocardial infarction, and stroke after matching by propensity score. The chronic periodontitis group were aged between 18 and 64 years old (126,975 individuals, about 75.5%). A total of 129,238 patients (76.8%) had a monthly income of less than NTD 25,000 Taiwanese dollars, and 104,740 individuals (62.3%) lived in urban areas (Table 1). Then, we calculated incidence densities and relative risks to gauge the correlation between chronic periodontitis and hematologic cancers (Table 2). The incidence density and 95% confidence interval for non-chronic periodontitis and chronic periodontitis were 0.19 (0.17–0.21) and 0.24 (0.21–0.26), respectively (Table 2).

In order to compare the cumulative incidence of hematological cancer in the chronic periodontitis and non-chronic periodontitis groups, the Kaplan–Meier method, with a log-rank test, was used for our analysis. As indicated in Table 3, the average follow-up duration in the non-chronic periodontitis and chronic periodontitis groups was 9.6 ± 2.8 years and 9.4 ± 2.8 years, respectively (determined by the log-rank test, *p* < 0.001). The mean interquartile ranges for hematologic cancers were 6.3 ± 3.4 years and 6.1 ± 3.2 years for the non-chronic periodontitis and chronic periodontitis groups, respectively (determined by the log-rank test, *p* = 0.636). As illustrated in Figure 2, the Kaplan–Meier plot for individuals with non-chronic periodontitis exhibited a notably lower cumulative incidence of hematologic cancer compared to those in the chronic periodontitis group during the 12-year follow-up period.

To assess the hazard ratio (HR) of hematologic cancers associated with the presence of non-chronic periodontitis and chronic periodontitis, we conducted a Cox proportional hazard regression analysis. Additionally, we utilized the adjusted hazard ratio (aHR) to represent a multivariate analysis that incorporates all variables used in this study. The results are shown in Table 4. In the univariate analysis, it was observed that the group with chronic periodontitis had a higher likelihood of developing hematologic cancers compared to the non-chronic periodontitis group (HR, 1.27; 95% CI: 1.09 to 1.48; *p* = 0.002), and the multivariate analysis, which included all variables considered in this study, confirmed that the chronic periodontitis group faced an elevated risk of developing hematologic cancers (aHR, 1.25; 95% CI: 1.07 to 1.45; *p* = 0.004). Moreover, our multivariate Cox proportional hazards model showed that being male (aHR = 1.21, 95% CI: 1.04 to 1.42; *p* = 0.014) and having hypertension (aHR = 1.34, 95% CI: 1.06 to 1.69; *p* = 0.015) were risk factors for hematologic cancers. It is worth mentioning that participants aged 18–64 years (aHR = 2.62, 95% CI: 1.77 to 3.89; *p* < 0.001) and those aged 65 years or older (aHR = 10.46, 95% CI: 6.87 to 15.91; *p* < 0.001) were also identified as being at higher risk of hematologic cancers.

In the subgroup multivariate analysis (Table 5), the occurrence of hematologic cancer was elevated in the chronic periodontitis group compared to the non-chronic periodontitis group, both in males and females. Notably, the aHR was higher for females than for males (1.30 and 1.25, respectively). Furthermore, chronic periodontal status significantly altered the association between individuals aged 18–64 years and their development of hematologic cancers (*p* for interaction = 0.012). Additionally, in the subtype multivariate analysis (Table 6), we categorized hematologic cancer into lymphoma and leukemia. The aHR for leukemia was 1.48 (95% CI: 1.13 to 1.93; *p* = 0.004), while for lymphoma it was 1.15 (95% CI: 0.96 to 1.37; *p* = 0.140) (Table 6).

## 4. Discussion

In summary of our comprehensive study on the association between chronic periodontitis and hematologic cancers using rigorous statistical analyses, the 12-year follow-up period revealed a significantly lower cumulative incidence of hematologic cancer in individuals without chronic periodontitis. Our Cox proportional hazard regression analysis identified being male and having hypertension as the significant risk factors for hematologic cancers. Moreover, our subtype analysis highlighted a higher adjusted hazard ratio for leukemia compared to lymphoma. These findings provide valuable insights into the nuanced relationship between chronic periodontitis and specific hematologic cancer subtypes. In addition, our study demonstrated the importance of gender and hypertension in understanding the complex interplay between periodontal health and hematologic malignancies.

The oral and periodontal microbiome might facilitate the development of cancer in distant areas through systemic inflammation, the indirect long-distance impact of virulence factors originating from the oral microbiota, and the direct migration of microorganisms via the bloodstream and oropharyngeal and respiratory pathways. Additionally, it could also affect patients’ response to treatments by interacting with the host immune response [23]. A previous review demonstrated that oral and esophageal cancers were consistent with the increased risk associated with periodontal disease [24]. Gastric and pancreatic cancers demonstrated various degrees of association with periodontal disease in most studies. However, lung, prostate, hematologic, and other cancers displayed either less-consistent associations with periodontal disease or a lack of sufficient studies to establish a predictable pattern [24]. Recently, Heikkilä et al. reported stronger associations of periodontitis with increased overall cancer mortality after adjustments for age, sex, socio-economic status, dental treatments, and diabetes [25].

Hematologic cancer, marked by the abnormal proliferation of undifferentiated white and red blood cells in the bone marrow, results in impaired cell function; as these undifferentiated cells (blasts) enter the bloodstream, they can infiltrate organs, including the oral cavity, leading to clinical manifestations like gingival bleeding and swelling, indicative of potential leukemia [26]. Previous studies have indicated that hematopoietic and lymphatic malignancies such as lymphomas and leukemia exhibited share risk factors like gender, immune dysregulation, older age, smoking, prior chemotherapy, and exposure to radiation [16,21,22]. Some studies have revealed that periodontal disease was associated with elevated risks of combined hematological cancers (HR, 1.18; 95% CI: 1.02 to 1.37) and lymphoid/hematopoietic malignancies in never-smokers (HR, 1.34; 95% CI: 1.08 to 1.67) [9,19]. Michaud et al. have reported that periodontal disease increased the risk of hematopoietic malignancies in male individuals (HR, 1.30; 95% CI: 1.11 to 1.53) [27]. Additionally, a report has suggested a higher association between periodontal disease and non-Hodgkin lymphomas (HR, 1.30; 95% CI: 1.11 to 1.51) [21].

This study was a cohort study aiming to investigate the association between chronic periodontitis and hematologic cancers in the Taiwanese population. The strength of this cohort design allows for the establishment of a cause relationship between chronic periodontitis and hematologic cancers. A 12-year follow-up period would enable researchers to observe the occurrence of these events. Through the use of the NHIRD, a large sample size, up to 601,628 patients, were recruited in this register-based cohort study. This could enhance its statistical power. With the adjustment of potential confounding variables, the results on the association between chronic periodontitis and hematologic cancers are more reliable and valid.

However, some possible limitations to the current study should be noted. First, the study extracted data from the NHIRD in Taiwan, while the diagnosis of chronic periodontitis was based on ICD-9 codes. However, there was no corresponding ICD-9 code for the new classification of periodontal diseases and conditions in 2017 [28]. The severity of the chronic periodontitis also could not be obtained from the NHIRD. Second, there is no record of oral habits such as alcohol consumption, betel nut chewing, cigarette smoking, lifestyle, dietary habits, and environmental factors in the NHIRD. These confounding variables may be unmeasured or unknown confounders that could impact the association between chronic periodontitis and hematologic cancers and should be underscored.

## 5. Conclusions

Through comprehensive statistical analyses using NHIRD, we found a higher hazard ratio for hematologic cancers in patients with chronic periodontitis. Furthermore, our study identified being male and having hypertension as the significant risk factors for hematologic cancers. The nuanced understanding gained from our investigation sheds light on the complex relationship between chronic periodontitis and specific subtypes of hematologic cancers. These findings could provide valuable insights for future research and clinical considerations.

## Figures and Tables

**Figure 1 cancers-16-01671-f001:**
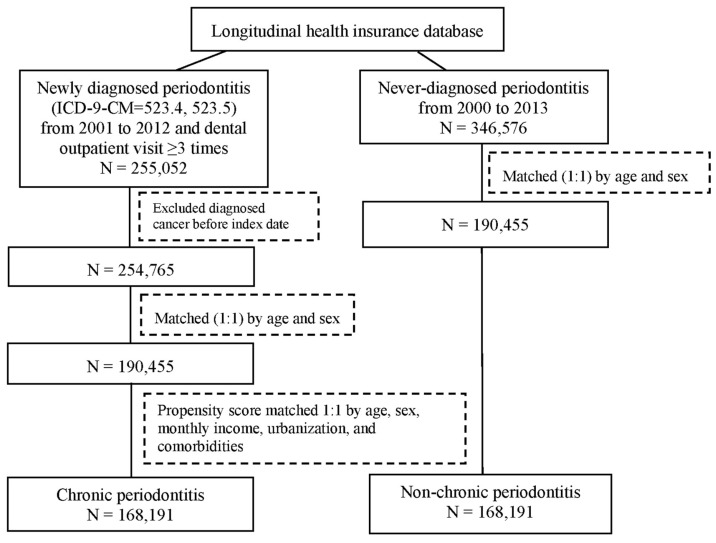
Flow chart of participant selection.

**Figure 2 cancers-16-01671-f002:**
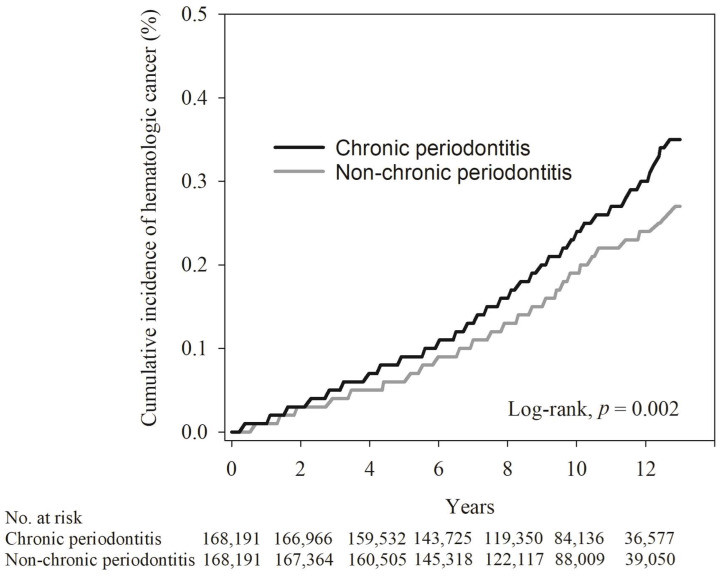
The Kaplan–Meier plot for the cumulative incidence of hematologic cancer in chronic periodontitis and non-chronic periodontitis subjects.

**Table 1 cancers-16-01671-t001:** Demographic characteristics of chronic periodontitis and non-chronic periodontitis groups.

	Before PSM			After PSM		
	Chronic Periodontitis (N = 190,455)	Non-Chronic Periodontitis (N = 190,455)	ASD *	Chronic Periodontitis (N = 168,191)	Non-Chronic Periodontitis (N = 168,191)	ASD *
Age			<0.001			0.020
<18	27,966 (14.7)	27,966 (14.7)		27,752 (16.5)	27,191 (16.2)	
18–64	146,987(77.2)	146,987(77.2)		126,975 (75.5)	128,304 (76.3)	
≥65	15,502 (8.1)	15,502 (8.1)		13,464 (8.0)	12,696 (7.5)	
Mean ± SD	36.9 ± 17.4	36.9 ± 17.4	<0.001	36.4 ± 17.3	35.9 ± 17.5	0.028
Sex			<0.001			0.007
Female	91,390 (48.0)	91,390 (48.0)		82,563 (49.1)	81,969 (48.7)	
Male	99,065 (52.0)	99,065 (52.0)		85,628 (50.9)	86,222 (51.3)	
Monthly income			0.248			0.009
<NTD 25,000	132,774 (69.7)	151,904 (79.8)		129,238 (76.8)	129,770 (77.2)	
NTD 25,000–NTD 40,000	23,046 (12.1)	18,783 (9.9)		18,720 (11.1)	18,653 (11.1)	
>NTD 40,000	34,635 (18.2)	19,768 (10.4)		20,233 (12.0)	19,768 (11.8)	
Urbanization			0.230			0.020
Urban	123,462 (64.8)	103,567 (54.4)		104,740 (62.3)	103,464 (61.5)	
Suburban	54,691 (28.7)	66,238 (34.8)		51,213 (30.4)	52,753 (31.4)	
Rural	12,302 (6.5)	20,650 (10.8)		12,238 (7.3)	11,974 (7.1)	
Hypertension	16,527 (8.7)	13,370 (7.0)	0.062	12,994 (7.7)	12,545 (7.5)	0.010
Hyperlipidemia	6721 (3.5)	4176 (2.2)	0.080	4199 (2.5)	4158 (2.5)	0.002
Diabetes	6959 (3.7)	5764 (3.0)	0.035	5499 (3.3)	5430 (3.2)	0.002
COPD **	3043 (1.6)	2161 (1.1)	0.040	2012 (1.2)	2102 (1.2)	0.005
Thyroid disease	2312 (1.2)	1415 (0.7)	0.048	1418 (0.8)	1412 (0.8)	<0.001
Asthma	2825 (1.5)	2171 (1.1)	0.030	2097 (1.2)	2097 (1.2)	<0.001
Myocardial infarction	5119 (2.7)	3519 (1.8)	0.056	3361 (2.0)	3411 (2.0)	0.002
Stroke	2411 (1.3)	2359 (1.2)	0.002	2078 (1.2)	2088 (1.2)	0.001
Insomnia	2940 (1.5)	1963 (1.0)	0.046	1933 (1.1)	1927 (1.1)	<0.001

PSM: propensity score matching; * ASD: absolute standardized difference; ** COPD: chronic obstructive pulmonary disease.

**Table 2 cancers-16-01671-t002:** Poisson regression of relative risk of chronic periodontitis group and non-chronic periodontitis groups.

	Non-Chronic Periodontitis	Chronic Periodontitis
N	168,191	168,191
Person years	1,581,487	1,605,135
No. of hematologic cancer cases	293	381
ID (95% C.I.)	0.19 (0.17–0.21)	0.24 (0.21–0.26)
Relative risk (95% C.I.)	Reference	1.28 (1.10–1.49)

ID: incidence density (per 1000 person years).

**Table 3 cancers-16-01671-t003:** The cumulative incidence of hematologic cancer in chronic periodontitis and non-chronic periodontitis subjects.

	Chronic Periodontitis (N = 168,191)	Non-Chronic Periodontitis (N = 168,191)	*p* Value
Follow-up duration (years)	9.6 ± 2.8	9.4 ± 2.8	<0.001
Time to hematologic cancer (years)	6.3 ± 3.4	6.1 ± 3.2	0.636

**Table 4 cancers-16-01671-t004:** Cox proportional hazard model analysis for risk of hematologic cancers.

	Univariate		Multivariate ^†^	
	HR (95% C.I.)	*p* Value	HR (95% C.I.)	*p* Value
Group				
Non-chronic periodontitis	Reference		Reference	
Chronic periodontitis	1.27 (1.09–1.48)	0.002	1.25 (1.07–1.45)	0.004
Age				
<18	Reference		Reference	
18–64	2.87 (1.95–4.24)	<0.001	2.62 (1.77–3.89)	<0.001
≥65	13.54 (9.06–20.23)	<0.001	10.46 (6.87–15.91)	<0.001
Sex				
Female	Reference		Reference	
Male	1.21 (1.04–1.41)	0.014	1.21 (1.04–1.42)	0.014
Monthly income				
<NTD 25,000	Reference		Reference	
NTD 25,000–NTD 40,000	0.80 (0.62–1.04)	0.096	0.96 (0.74–1.26)	0.793
>NTD 40,000	1.07 (0.86–1.33)	0.561	1.25 (0.99–1.58)	0.058
Urbanization				
Urban	Reference		Reference	
Suburban	1.04 (0.88–1.22)	0.676	0.96 (0.82–1.14)	0.672
Rural	1.04 (0.77–1.40)	0.801	0.81 (0.60–1.09)	0.162
Hypertension	3.25 (2.70–3.93)	<0.001	1.34 (1.06–1.69)	0.015
Hyperlipidemia	2.88 (2.10–3.93)	<0.001	1.31 (0.94–1.85)	0.114
Diabetes	2.75 (2.08–3.63)	<0.001	1.18 (0.87–1.60)	0.294
COPD **	2.82 (1.86–4.27)	<0.001	1.10 (0.71–1.70)	0.670
Thyroid disease	1.88 (1.04–3.41)	0.038	1.55 (0.85–2.83)	0.151
Asthma	2.09 (1.29–3.39)	0.003	1.25 (0.76–2.06)	0.372
Myocardial infarction	3.79 (2.83–5.09)	<0.001	1.36 (0.98–1.88)	0.065
Stroke	2.78 (1.80–4.29)	<0.001	0.89 (0.56–1.39)	0.600
Insomnia	2.55 (1.60–4.08)	<0.001	1.32 (0.82–2.12)	0.252

** COPD: chronic obstructive pulmonary disease. ^†^ Adjusted for age, sex, monthly income, urbanization, and comorbidities.

**Table 5 cancers-16-01671-t005:** Subgroup analysis for risk of hematologic cancer.

	Chronic Periodontitis		Non-Chronic Periodontitis		aHR (95% C.I.)	*p* Value
	N	No. of Hematologic Cancer	N	No. of Hematologic Cancer		
Age						
<18 ^a^	30,327	281	30,324	246	0.55 (0.25–1.20)	0.130
18–64 ^b^	33,416	344	33,600	326	1.49 (1.23–1.80)	<0.001
≥65 ^b^	6728	45	6547	32	1.00 (0.75–1.31)	0.977
					*p* for interaction =	0.012
Sex ^c^						
Female	37,004	397	36,999	376	1.30 (1.03–1.63)	0.026
Male	33,467	273	33,472	228	1.25 (1.02–1.53)	0.033
					*p* for interaction =	0.810

aHR: adjusted HR. ^a^: adjusted for age, sex, and urbanization. ^b^: adjusted for age, sex, monthly income, urbanization, and comorbidities. ^c^: adjusted for age, monthly income, urbanization, and comorbidities.

**Table 6 cancers-16-01671-t006:** Subtype analysis for risk of hematologic cancer.

	Chronic Periodontitis		Non-Chronic Periodontitis		aHR ^†^ (95% C.I.)	*p* Value
	N	No. of Event	N	No. of Event		
Hematologic cancer						
Lymphoma	168,191	256	168,191	214	1.15 (0.96–1.37)	0.140
Leukemia	168,191	136	168,191	89	1.48 (1.13–1.93)	0.004

aHR: adjusted HR. ^†^ Adjusted for age, sex, monthly income, urbanization, and comorbidities.

## Data Availability

Restrictions apply to the availability of these data. Data was obtained from National Health Insurance database and are available from the authors with the permission of National Health Insurance Administration of Taiwan.

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
