# Peer review of "Association between Periodontitis and Hematologic Cancer: An NHIRD Cohort Study in Taiwan"

_cancers, 2024, doi:10.3390/cancers16091671_

Round 1

Reviewer 1 Report

Comments and Suggestions for Authors

My opinion:

1. Male gender and hypertension were identified as significant risk factors for haematological malignancies.

2. The association between chronic periodontitis and specific subtypes of haematologic cancers was confirmed.

Author Response

  1. Male gender and hypertension were identified as significant risk factors for haematological malignancies.

Reply: Thanks for your insightful feedback. We have added the text to the conclusion of the article.

  1. The association between chronic periodontitis and specific subtypes of haematologic cancers was confirmed.

Reply: Thanks for your valuable feedback. We have incorporated the text into the conclusion of the article.

Reviewer 2 Report

Comments and Suggestions for Authors

Thank you very much for the opportunity to review this cross-sectional and a little bit longitudinal study.

The only main problem of the paper is the missing actual classification of periodontal disease and periodontal health (Tonetti et al 2018). The discussion on literature should be analysed and discussed and analysed for different stages 1-4 of periodontitis in the studies. Some shortenings are possible.

As a european I can not understand proposed racial differences presented in the discussion.

Author Response

 Thanks for taking the time to review our study and provide the article (Tonetti et al., 2018) as a reference for periodontal disease classification. Regarding the classification of periodontal disease and periodontal health, we identified periodontitis based on ICD-9-CM diagnostic codes 523.4 and 523.5. However, it should be noted that ICD-9-CM diagnostic codes may not directly correspond to the classification of periodontal disease due to limitations in the discussion. We have also regarded the Tonetti et al., 2018 article as reference number 28.

Reviewer 3 Report

Comments and Suggestions for Authors

Minor revision.

English check.

Add citation Heikkilä et al Int J Cancer -18,

and cite accordingly.

Comments on the Quality of English Language

Same as above.

After minor revision ok for publication.

Author Response

 Thank you for your suggestion. We have added the article by Heikkilä et al Int J Cancer -18 into our article's discussion sections as a reference (number 25).

Reviewer 4 Report

Comments and Suggestions for Authors

The manuscript, authored by Huang LG, is interesting and well-done. The authors evaluated several variables related to chronic periodontitis and hematological disease. Despite the interesting study, the authors need to address minor comments.

Firstly, the authors describe the Asian population in the study, but only study Taiwanese individuals. Table 1 does not correctly define the abbreviation PSM.

The authors describe that gender and hypertension were the most frequent risk factors related to hematologic cancer in chronic periodontitis, but these variables were not properly addressed in the discussion. I suggest making hypotheses associated with hematologic cancer, periodontitis, and hypertension. What type of hematologic cancer was most related to hypertension, if hypertension was associated with hematologic cancer, or if hematological cancer was associated with hypertension?

The manuscript is interesting and provides clinical risk factors related to periodontitis and hematological cancer, but the discussion needs to be strengthened by making hypotheses associated with the described variables.

Author Response

Thanks for your kind correction. The title has been revised as “Asssociation between periodontitis and hematologic cancer: A NHIRD corhort study in Taiwan”. The full name of PSM has been added at the bottom of Table 1. The association between chronic periodontitis and specific subtypes of hematologic cancers has been confirmed. In addition, the multivariate analysis, categorizing hematologic cancer into lymphoma and leukemia subtypes, revealed that the type of leukemia was most strongly associated with hypertension (Table 6).

Round 2

Reviewer 2 Report

Comments and Suggestions for Authors

Thank you for your review report reply. All necessary changes in the manuscript are adaequat and sufficient.